# A comprehensive wireless neurological and cardiopulmonary monitoring platform for pediatrics

Jeremy N. Wong[1,2], Jessica R. Walter[3ʘ], Erin C. Conrad[4ʘ], Dhruv R. Seshadri[5ʘ], Jong Yoon Lee[5], Husein Gonzalez[5], William Reuther[5], Sue J. Hong[2,6,7], Nicolò Pini[8,9], Lauren Marsillio[6,7†], Khrystyna Moskalyk[1], Mariana Vicenteno[1], Erik Padilla[1], Olivia Gann[5], Ha Uk Chung[5], Dennis Ryu[5], Carlie du Plessis[10], Hein J. Odendaal[10], William P. Fifer[8,9,11], Joyce Y. Wu[1,2], Shuai Xu[5,12,13,14]*

1 Epilepsy Center, Division of Pediatric Neurology, Lurie Children's Hospital of Chicago, Chicago, Illinois, United States of America, 2 Department of Pediatrics, Division of Neurology, Northwestern University Feinberg School of Medicine, Chicago, Illinois, United States of America, 3 Department of Obstetrics and Gynecology, Northwestern University Feinberg School of Medicine, Chicago, Illinois, United States of America, 4 Department of Neurology, University of Pennsylvania, Philadelphia, Pennsylvania, United States of America, 5 Sibel Inc., Niles, Illinois, United States of America, 6 Division of Critical Care, Lurie Children's Hospital of Chicago, Chicago, Illinois, United States of America, 7 Department of Pediatrics, Division of Critical Care, Northwestern University Feinberg School of Medicine, Chicago, Illinois, United States of America, 8 Department of Psychiatry, Columbia University Irving Medical Center, New York, New York, United States of America, 9 Division of Developmental Neuroscience, New York State Psychiatric Institute, New York, New York, United States of America, 10 Department of Obstetrics and Gynecology, Faculty of Medicine and Health Sciences, Stellenbosch University, Cape Town, South Africa, 11 Department of Pediatrics, Columbia University Irving Medical Center, New York, New York, United States of America, 12 Simpson Querrey Institute, Northwestern University, Chicago, Illinois, United States of America, 13 Department of Biomedical Engineering, Northwestern University, Evanston, Illinois, United States of America, 14 Department of Dermatology, Northwestern University Feinberg School of Medicine, Chicago, Illinois, United States of America

ʘ These authors contributed equally to this work.
† Deceased.
* steve.xu@sibelhealth.com

**Data Availability Statement:** Data is provided as Supporting Information file.

**Funding:** JNW, WPF, SJH and SX recognize funding from the Bill and Melinda Gates Foundation

## Abstract

Neurodevelopment in the first 10 years of life is a critical time window during which milestones that define an individual's functional potential are achieved. Comprehensive multimodal neurodevelopmental monitoring is particularly crucial for socioeconomically disadvantaged, marginalized, historically underserved and underrepresented communities as well as medically underserved areas. Solutions designed for use outside the traditional clinical environment represent an opportunity for addressing such health inequalities. In this work, we present an experimental platform, ANNE EEG, which adds 16-channel cerebral activity monitoring to the existing, USA FDA-cleared ANNE wireless monitoring platform which provides continuous electrocardiography, respiratory rate, pulse oximetry, motion, and temperature measurements. The system features low-cost consumables, real-time control and streaming with widely available mobile devices, and fully wearable operation to allow a child to remain in their naturalistic environment. This multi-center pilot study successfully collected ANNE EEG recordings from 91 neonatal and pediatric patients at academic quaternary pediatric care centers and in LMIC settings. We demonstrate the

(INV-019423). The funders had no role in study design, data collection and analysis, decision to publish, or preparation of the manuscript.

**Competing interests:** I have read the journal's policy and the authors of this manuscript have the following competing interests: JRW reports a spouse with stock options in a private commercial entity commercializing the technology, and a royalty interest to patents related to the technology. SX, JYL, HG, DR, WR, DS, HUC, and OG are all employees with stock ownership of a private commercial entity commercializing the technology. EC receives consulting income from Epiminder, an EEG device company.

practicality and feasibility to conduct electroencephalography studies with high levels of accuracy, validated via both quantitative and qualitative metrics, compared against gold standard systems. An overwhelming majority of parents surveyed during studies indicated not only an overall preference for the wireless system, but also that its use would improve their children's physical and emotional health. Our findings demonstrate the potential for the ANNE system to perform multimodal monitoring to screen for a variety of neurologic diseases that have the potential to negatively impact neurodevelopment.

## Author summary

Neurodevelopment refers to an ensemble of multifaceted processes by which humans develop the neural pathways underlying age-appropriate functional potential. Monitoring cerebral activity via scalp electroencephalography (EEG) can serve as a noninvasive and sensitive tool to probe brain functioning and inform measures of neurodevelopment as well as evaluate various disabilities and neurologic disease such as epilepsy–which affects millions of children with a disproportionate prevalence in developing countries. At present there is a lack of neurodevelopmental monitoring systems that feature both cerebral and cardiopulmonary monitoring. We present a novel wireless monitoring system that adds EEG measurements to an existing USA FDA-cleared wearable sensors platform that measures cardiac activity, blood oxygenation, motion, and skin temperature. In this paper, we demonstrate the system's ability to record high-quality cerebral neonatal and pediatric activity in a variety of clinical and research environments, including in a low-middle income country (LMIC) setting. Moreover, this system paired with limb and chest sensors, has the potential to perform clinical-grade comprehensive neurodevelopmental monitoring in a child's naturalistic environment. Potential use cases range from evaluating for neurologic disease at a medical clinic to brief at-home neurodevelopmental screening.

## Introduction

Neurodevelopment from birth to pre-adolescence is a sensitive period for attaining functional and cognitive milestones. During this timeframe, cerebral volume increases four-fold [1] to support structural and functional development occurring via various mechanisms such as neurogenesis, pruning, and myelination [2,3]. The result of such processes enables the achievement of behavioral and cognitive milestones as well as downstream effects on metabolism and organ development. Neurodevelopment can be adversely impacted by neurologic diseases such as birth trauma leading to hypoxic ischemic encephalopathy, perinatal stroke, and epilepsy. In particular, the latter is the most common chronic neurologic disorder affecting over 50 million people worldwide, over 40 million of whom live in developing countries and low-middle income settings [4]. Up to 4% of all children have epilepsy, with the highest prevalence in rural areas and developing countries [5]. Prompt seizure detection in infancy and childhood is critical as untreated ongoing seizures correlate with adverse neurodevelopmental outcomes [6].

   While there are existing systems for objectively tracking neurodevelopment in terms of structural and functional brain development, they all have significant limitations. Magnetic resonance imaging (MRI) is used to measure cerebral structure and volume, both of which are

hypothesized to be key markers of neurodevelopment [7]. However, there are numerous limitations for their widespread and regular use, including their extremely high cost of approximately $1 million USD / Tesla [8] and specialized infrastructure needs for super-conducting magnets. As a result, their prevalence per capita in low and middle income countries (LMICs) are less than 1/50[th] than that of the US [9]. In recent years, more inexpensive and easier to maintain low field systems [10] are more promising for LMIC adoption, though their uses to date have been largely experimental [8,11] or proof-of-concept [10]. Electroencephalography (EEG) has recently been shown to aid in measuring cerebral function in individuals with normal as well as adverse neurodevelopmental trajectories [12]. Specifically, multidimensional features extracted from EEG have been shown to be sensitive markers for the adverse effects of *in-utero* exposures such as prenatal maternal smoking [13] and alcohol use [14]. EEG is also the gold standard in quantifying sleep cycles, which is a key marker of brain development in preterm and term neonates and potentially across the lifespan [15]. However, current standard-of-care EEG monitoring systems are often not portable as they consist of wired attachments to a base headbox unit and large bulky monitors. These cumbersome connections often limit patient mobility and visibility by caregivers, impede clinical care, and limits the use of such systems in naturalistic environments. There has been a recent emergence of wireless EEG devices such as Rapid Response EEG (Ceribell, Palo Alto USA) for adults and Epilog (Epitel, Salt Lake City USA) for research purposes, each with early experimental data [2,16–18]. However, these have a sub-standard number of electrodes (up to 10) and were not designed to meet the unique anatomic and physiological considerations of neonates and children. Furthermore, none meet current American Clinical Neurophysiology Society (ACNS) clinical guidelines for neonate and infant cerebral monitoring, which recommend integration of non-EEG measurements such as electrocardiography (EKG) and respiration [19]. Lastly, none of these systems integrate with other cardiopulmonary and positional monitoring tools that provide key physiological data to inform neurodevelopment.

There is an urgent need for advanced integrated mobile/portable systems that allows for comprehensive, time synchronized multimodal measurements of neurodevelopment and brain health, designed for use both in the pediatric population and amenable to LMIC settings. To address these unmet needs, we present an experimental platform, ANNE EEG, which adds 16 channel high quality cerebral activity monitoring to the existing, FDA-cleared ANNE wireless monitoring platform which provides continuous EKG, respiratory rate, pulse oximetry, motion, and temperature measurements [20]. This allows for the unified collection and storage of multimodal physiologic data, allowing for the potential of advanced analytics to predict neurodevelopment.

## Results

### Neurodevelopment system design and engineering

Fig 1 displays key aspects of the system's sensor design. A four-layer flexible printed circuit board is fabricated for the ANNE EEG sensor that consists of a stack-up of copper, polyimide, coverlay, and soldermask. The sensor consists of the following; a system-on-a-chip (SoC, ISP1807, Insight SIP); high-precision and low noise analog-front-ends (AFE) for a maximum of 16 channels; and a power management unit for wireless charging and supplying power to the on-board electronics with a rechargeable Li-polymer battery (230 mAh). The SoC controls the two AFE with on-board low-noise PGAs and sigma delta ADCs to collect clinical grade EEG data, where each channel samples at 1 kHz with 24-bit resolution and 0.28 μVrms input noise. The sensor board is enclosed with a ruggedized plastic housing made of Acrylonitrile butadiene styrene (ABS) with 16 channel connectors to standard EEG electrodes.

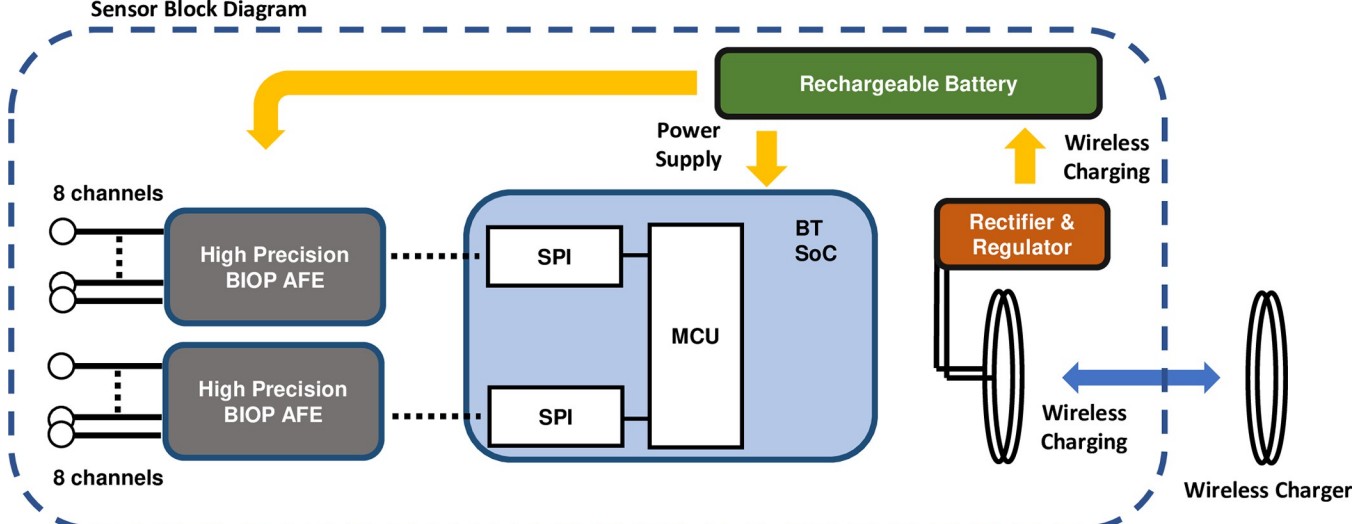

**Fig 1. Sensor block diagram of neurodevelopmental system.** The sensor consists of a system-on-a-chip (SoC, ISP1807, Insight SIP), high-precision and low noise analog-front-ends (AFE) for 16 channels, power management unit for wireless charging and supplying power to the on-board electronics with a 230mAh rechargeable Li-polymer battery. The SoC controls the two AFE with on-board low-noise PGAs and sigma delta ADCs to collect clinical grade EEG data, where each channel samples at 1 kHz with 24-bit resolution.

Panel A of Fig 2 provides an exploded schematic view of the device to highlight the underlying technology. ANNE EEG has a battery life of 8 hours between charges and a projected cost of $1,000 USD to allow for cost-effective operation in LMIC environments. The device is fully reusable. The electronics are enclosed with a ruggedized plastic housing, which supports up to 16 channels through standard clinical-grade EEG connectors. The housing also can be configured to have a neck carrying device by connecting a strap to the housing. The wireless charger is designed to charge up to two sensors at once through an inductive coupling at 13.56 MHz. In addition, the sensor can support a time-synchronized wireless body sensor network with ANNE Chest and ANNE Limb sensors for comprehensive monitoring.

Panel B of Fig 2 shows a schematic view of the system's chest (top) and limb (bottom) modules. The chest module measures EKG, cardiac seismocardiogram (SCG), chest wall movement for respiratory rate (RR), and skin temperature, sampled at 512, 416, 52, and 0.2 Hz, respectively. Measurements are made via a biopotential AFE, a high-frequency three-axis inertial measurement unit (IMU), and a clinical-grade thermometer. The limb sensor measures photoplethysmogram (PPG) and skin temperature, sampled at 256 and 0.2 Hz, respectively. These are recorded via an integrated pulse oximetry module and a clinical-grade thermometer. Red and infrared light-emitting diodes are employed to measure peripheral capillary oxygen saturation ($SpO_2$).

Fig 3A provides a comprehensive view of multimodal wireless vital sign measurements on a patient. The chest, limb, and EEG units provide a wide variety of both directly measured and derived clinical data. The EEG unit, via electroencephalography, can assess cerebral activity, provide sleep staging, and detect epileptiform discharges. The chest unit detects skin and body temperature, gathers electrocardiography data such as heart rate and bioimpedance, and senses chest wall movement. The limb unit sensor provides peripheral oximetry and peripheral arterial tonometry. This data can be time synchronized as depicted in Fig 2B to provide comprehensive multimodal monitoring. For instance, the presence of a sleep spindle on EEG supports that a child is in stage II sleep, which can be supported by concomitant measurements of heart rate, respiratory rate, and limb movements.

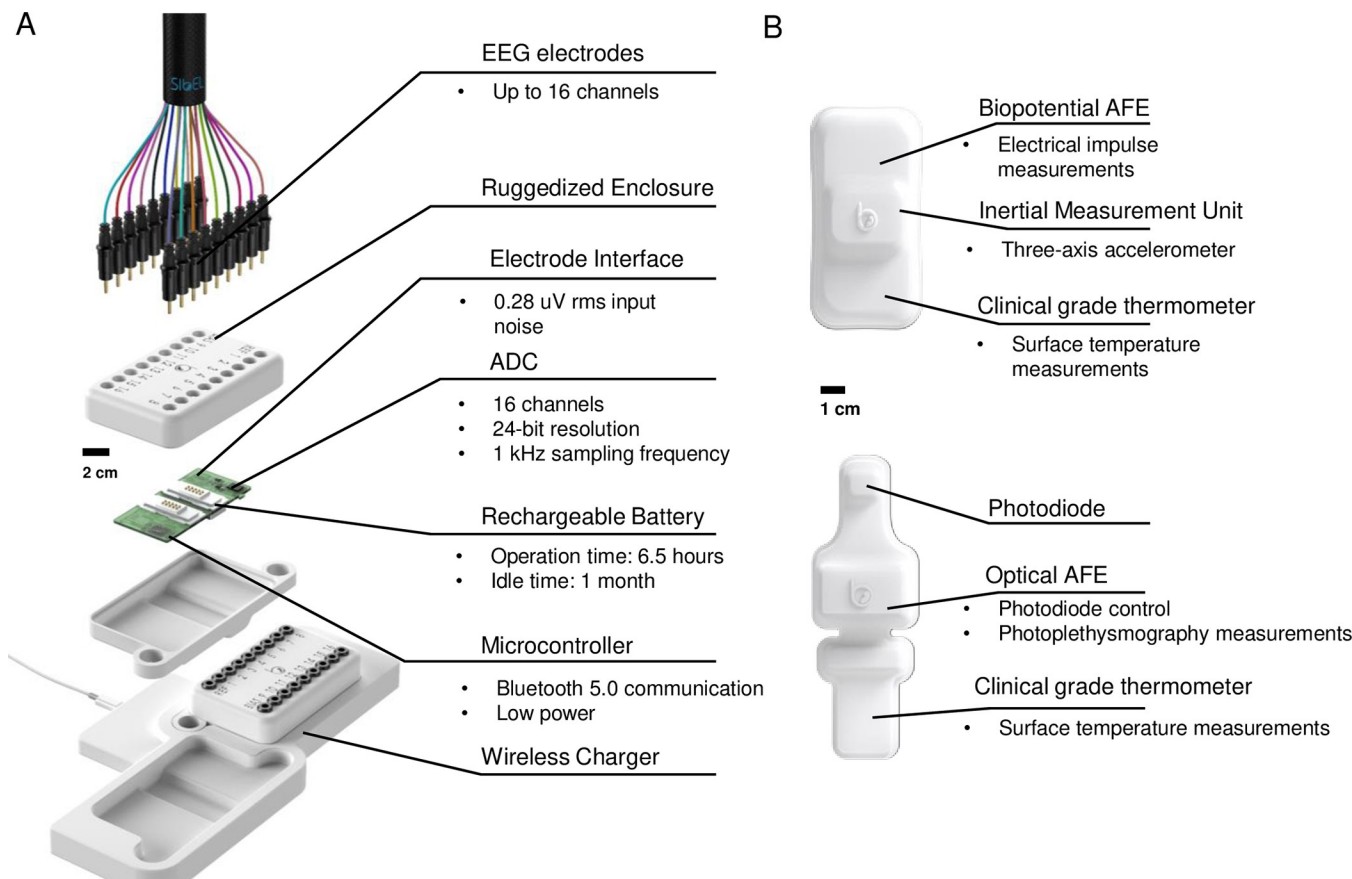

**Fig 2. Design and mechanical characterization of a wireless neurodevelopmental system for infants and children.** (A) Schematic illustration and exploded view of EEG module. (B) Schematic illustration of chest (top) and limb (bottom) devices.

## Neurodevelopment system qualitative and quantitative data analysis

EEG data from a total of 91 subjects across multiple centers was analyzed (see: Materials and Methods for study design and data collection). A bandpass filter of 1-70Hz with a notch filter of 60Hz was applied for all EEG visualization figures in accordance with clinical standards [21]. Fig 4 displays typical graphoelements of neonatal sleep activity measured by ANNE EEG. Analysis of quiet and active stages of neonatal sleep has been shown to predict neurodevelopmental outcome [22]. Fig 4A displays features of quiet sleep obtained from recordings collected in the HIC setting, and Fig 4B displays separate features identified in active sleep traces, both acquired in the LMIC setting.

Fig 5 displays characteristic outputs of cerebral activity as concurrently measured both by ANNE EEG and a clinical gold-standard EEG monitoring system. An atypical absence seizure from a study patient is visualized both from a clinical standard longitudinal bipolar visualization [23] of two representative channels (Fig 5A) and a longitudinal bipolar visualization of all sixteen channels (Fig 5B). Data from each system was post-processed with a 1–70 Hz bandpass filter with 60 Hz notch filter, which is standard clinical practice for EEG interpretation [24,25]. Both systems visualized 2.5–3.0 Hz stereotyped diffuse spike-wave activity with abrupt onset and offset, all of which are key features of this seizure type. EEG visualization of the seizure from each system was indistinguishable from one another.

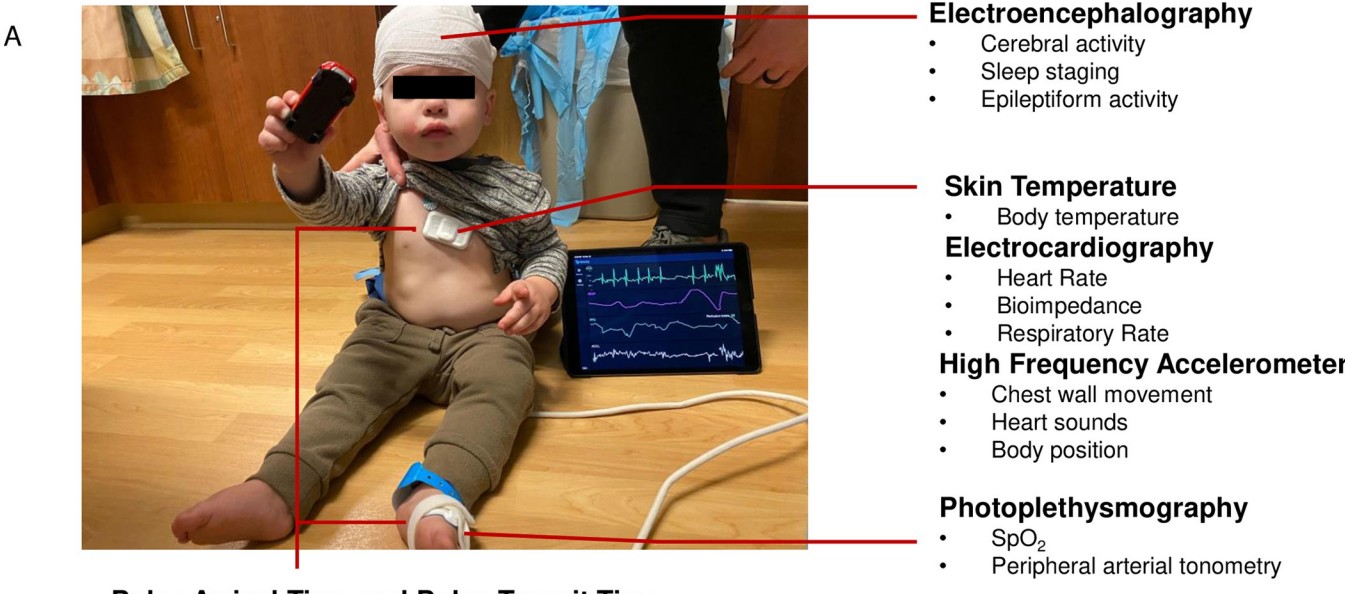

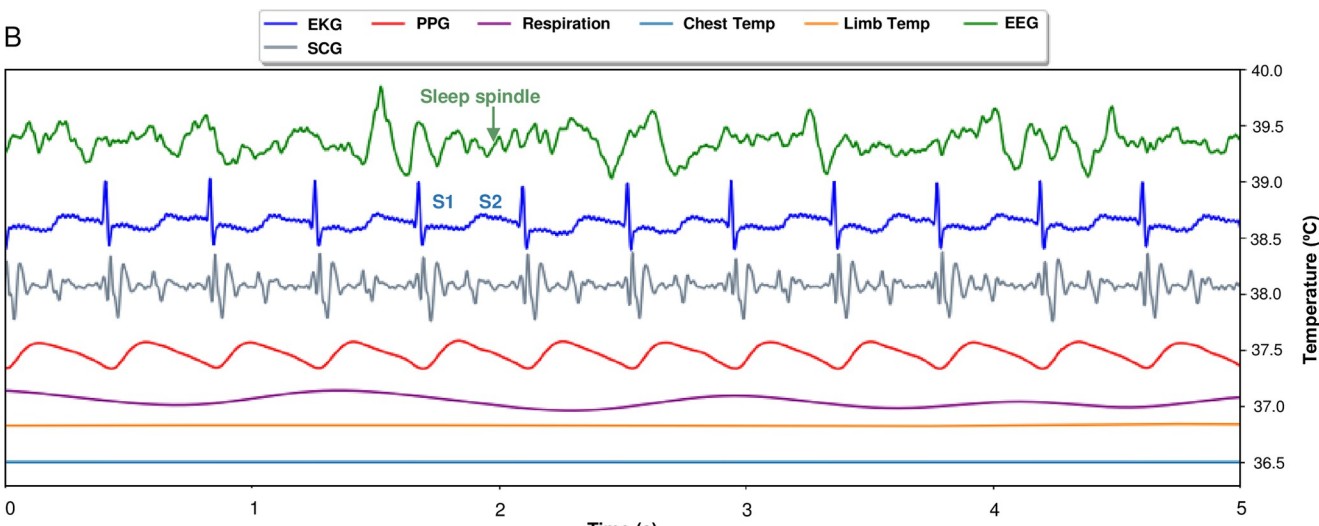

**Fig 3. Continuous, wireless, and real-time neurological and cardiopulmonary monitoring via a time-synchronized sensor network.** (A) Photograph of ANNE EEG, chest and limb units on a 6 year old child with measured and derived metrics. (B) Visual representation of multimodal vital sign measurements. Multimodal data from ANNE EEG, chest and limb devices; Electroencephalography (EEG), electrocardiography (EKG), photoplethysmography (PPG), respiration, chest temperature, limb temperature, and seismography (SCG) over a 5 second interval. Sleep spindles are a marker of stage 2 sleep. S1 and S2 are heart sounds indicating systole and diastole, respectively.

On visual inspection, concurrent outputs between the two systems were similar for all patients, independent of their EEG findings. However, a few differences were noted. The ANNE EEG system's outputs displayed a slightly sharper contour to their waveforms, likely owing to its higher sampling frequency (1000 Hz) vs that of the gold-standard system (512 Hz). In addition, at times there was some discordance in either phase or amplitude, likely due to the gold standard and ANNE EEG electrodes being separate and spatially separated by

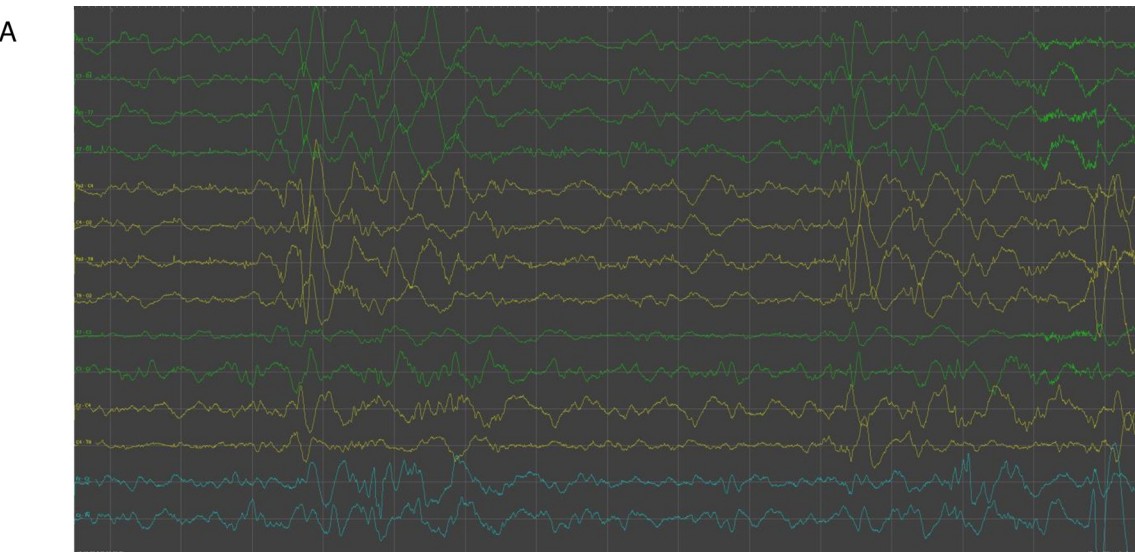

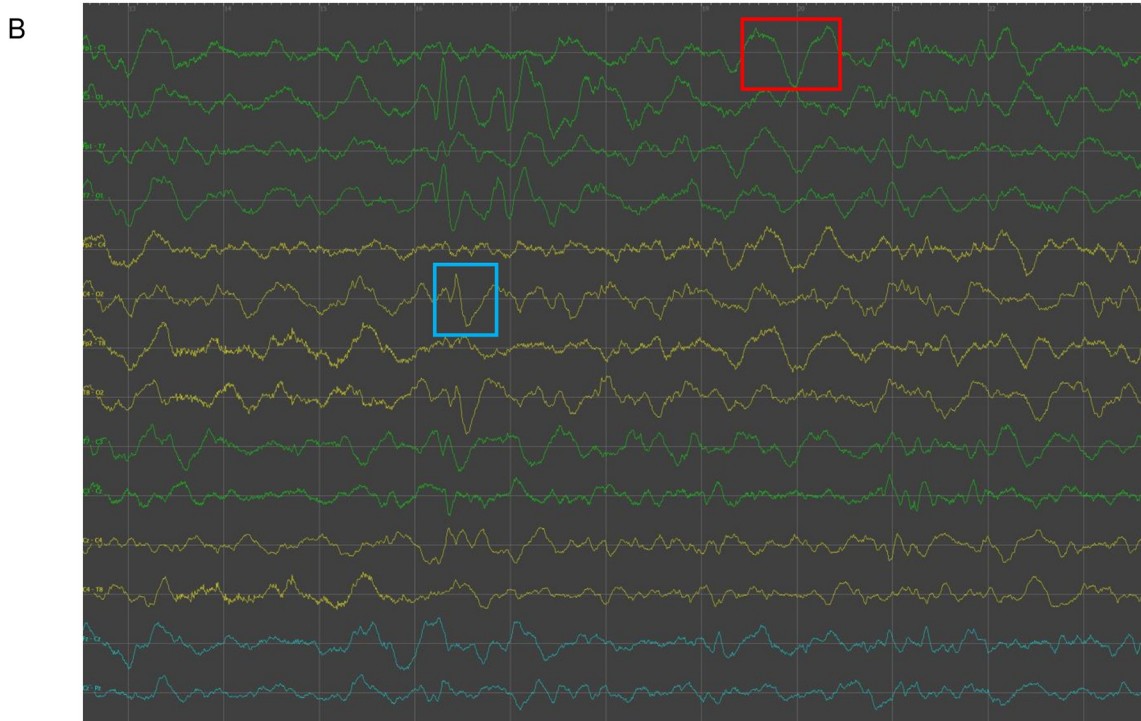

**Fig 4. Neonatal sleep with characteristic EEG graphoelements.** (A) Quiet sleep obtained from a participant at Columbia University Hospital (New York, USA). This snapshot displays a tracé alternant pattern, defined by alternating periods of short-duration, high-voltage activity separated by bursts of < 50uV activity lasting up to 5 seconds (B) Active sleep obtained from a participant at Stellenbosch University (Stellenbosch, South Africa) in the LMIC setting. Anterior dysrhythmias (blue box) are 50–100 uV sharp waves lasting up to 200ms seen over the frontal regions. Enchoches frontales (red box) are 50-100uV broad diphasic transients, also seen in active sleep.

approximately 1cm. The electrode differences resulted in disparities likely due to single electrode artifact, variations in skin-electrode adhesion, or nonuniform effects of head movement artifact.

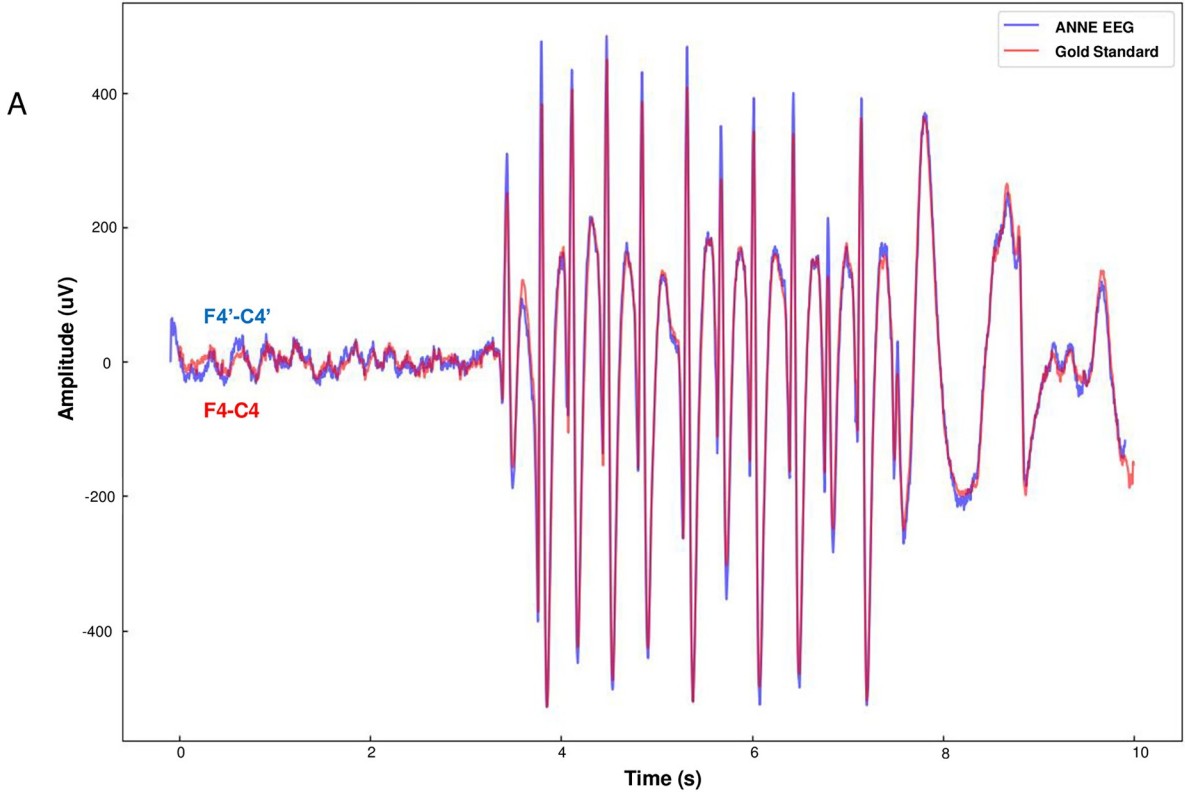

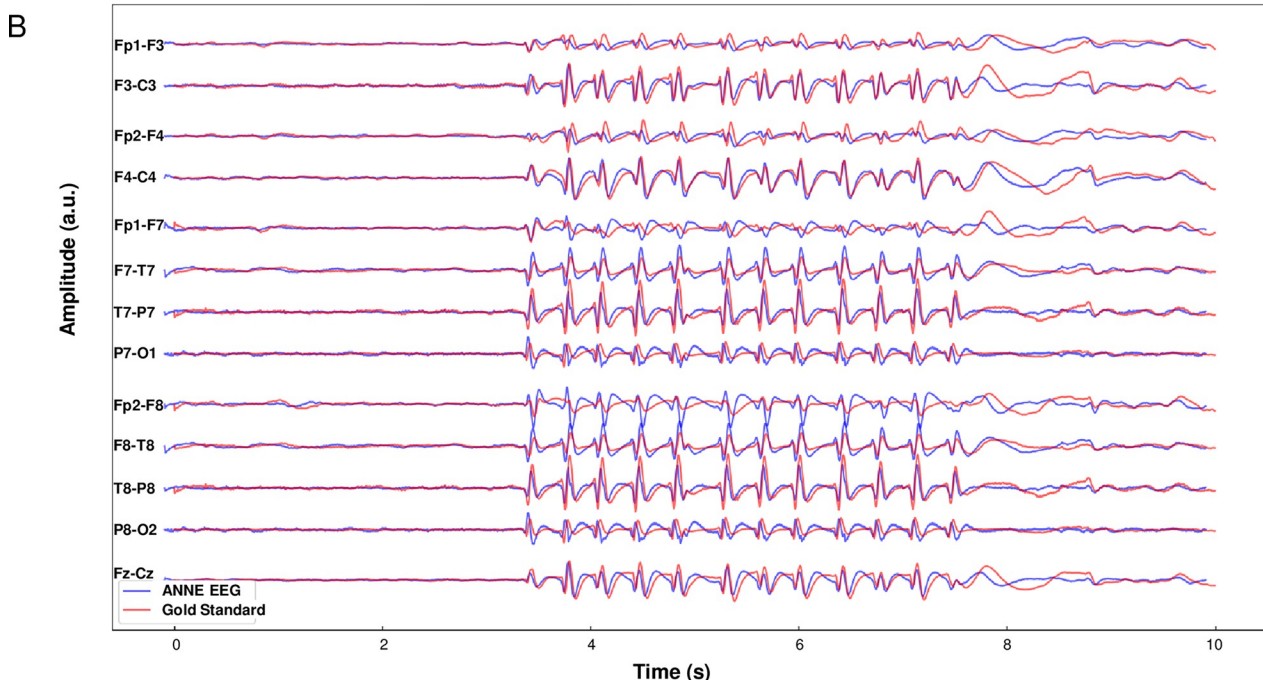

**Fig 5. Visual EEG comparison to gold-standard recording system.** (A) Comparison of absence seizure between F4-C4 and F4'-C4' channels between gold standard and ANNE EEG, respectively (B) 16 channel comparison of absence seizure between gold standard and ANNE EEG.

Quantitative data analysis was performed for the 30 subjects that comprised the validation study cohort. Similar analysis for the other 61 subjects in the feasibility arm could not be performed due to the lack of varied ages and control EEG data. For patients who fell asleep during EEG recording or had epileptiform discharges, 30 minute epochs were identified via manual inspection that had either continuous sleep or epileptiform discharges by a board-certified Child Neurologist (J.N.W.). Specifically, two quantitative EEG measures, alpha-delta ratio, and spike count were performed. Alpha-delta ratio is a validated marker of sleep [41,42]. Epileptiform spikes are a biomarker of seizures and epilepsy [43,44]. Automated analysis for both metrics were performed via a previously validated software package using MATLAB (Mathworks, California USA) software [45].

Relative power spectrum analysis was calculated for each channel in the validation study cohort. Sleep epochs were chosen to minimize the effects of movement artifact. The same representative 30 minute epochs of continuous sleep were used as in the alpha-delta ratio calculations. Power spectra for each of the 16 channels were divided into the following physiologically relevant frequencies; delta (0.5 to 4 Hz), theta (4 to 8 Hz), alpha (8 to 13 Hz), beta (13 to 30 Hz), and gamma (30Hz to 50Hz). Each of these frequency bands were then averaged across all channels along with the total power for the epoch (Table 1). Boxplots were then generated to compare relative delta ratio and total power between the subgroup with epileptiform discharges versus the subgroup without epileptiform discharges.

Fig 6 displays validation analyses via Bland-Altman and R-squared scatterplots (see: Methods for statistical details). We demonstrate strong concordance both for alpha-delta ratio (Fig 6A and 6B) as well as spike count (Fig 6C and 6D). Bland Altman analysis for both metrics demonstrated mean differences of very near zero and largely symmetric 95% confidence intervals. R-squared scatterplots displayed alpha-delta ratio and spike count $R^2$ coefficients of 0.98 and 0.99 respectively.

Fig 7 compares the distribution of EEG power (absolute broadband power and relative delta frequency) in sleep without epileptiform spikes versus sleep with epileptiform spikes.

**Table 1. Results of relative power spectrum for each frequency band in validation study.**

| Subject ID | Age in years | Delta | Theta | Alpha | Beta | Gamma | Total Power in uV$^2$ |
|---|---|---|---|---|---|---|---|
| | | | Sleep without epileptiform spikes | | | | |
| LCH-P001 | 0.5 | 0.75953 | 0.19845 | 0.02397 | 0.01277 | 0.00528 | 4.10834E+08 |
| LCH-P005 | 1.5 | 0.91923 | 0.06977 | 0.00645 | 0.00380 | 0.00076 | 2.93549E+09 |
| LCH-P007 | 10 | 0.64957 | 0.18292 | 0.09416 | 0.06202 | 0.01134 | 1.80218E+08 |
| LCH-P008 | 17 | 0.48838 | 0.17911 | 0.17572 | 0.11038 | 0.04641 | 8.98731E+07 |
| LCH-P009 | 14 | 0.75224 | 0.19950 | 0.03511 | 0.01161 | 0.00154 | 1.89937E+08 |
| LCH-P010 | 9 | 0.54931 | 0.35681 | 0.04303 | 0.03898 | 0.01187 | 4.02739E+08 |
| LCH-P013 | 11 | 0.56072 | 0.27892 | 0.09180 | 0.05916 | 0.00940 | 1.13181E+08 |
| LCH-P016 | 4 | 0.85477 | 0.10433 | 0.02124 | 0.01785 | 0.00180 | 7.25192E+08 |
| LCH-P017 | 15 | 0.49126 | 0.25712 | 0.16780 | 0.07438 | 0.00944 | 7.20335E+07 |
| LCH-P018 | 10 | 0.74486 | 0.17667 | 0.05171 | 0.02524 | 0.00152 | 5.21557E+08 |
| LCH-P019 | 14 | 0.68532 | 0.20421 | 0.07152 | 0.03695 | 0.00200 | 2.91378E+08 |
| LCH-P021 | 18 | 0.50284 | 0.18431 | 0.23353 | 0.06997 | 0.00935 | 8.44352E+07 |
| LCH-P027 | 1.5 | 0.75968 | 0.19476 | 0.03036 | 0.01296 | 0.00224 | 7.38864E+08 |
| LCH-P029 | 0.75 | 0.86955 | 0.10952 | 0.01558 | 0.00479 | 0.00055 | 2.62775E+09 |
| | | | Sleep with epileptiform spikes | | | | |
| LCH-P024 | 6 | 0.71193 | 0.19078 | 0.06037 | 0.03317 | 0.00375 | 5.50290E+08 |
| LCH-P025 | 4 | 0.85876 | 0.11556 | 0.01868 | 0.00647 | 0.00054 | 1.16730E+09 |
| LCH-P026 | 6 | 0.78500 | 0.17510 | 0.02985 | 0.00936 | 0.00069 | 1.95387E+09 |
| LCH-P030 | 1.5 | 0.76849 | 0.14070 | 0.05233 | 0.02963 | 0.00885 | 6.49197E+08 |

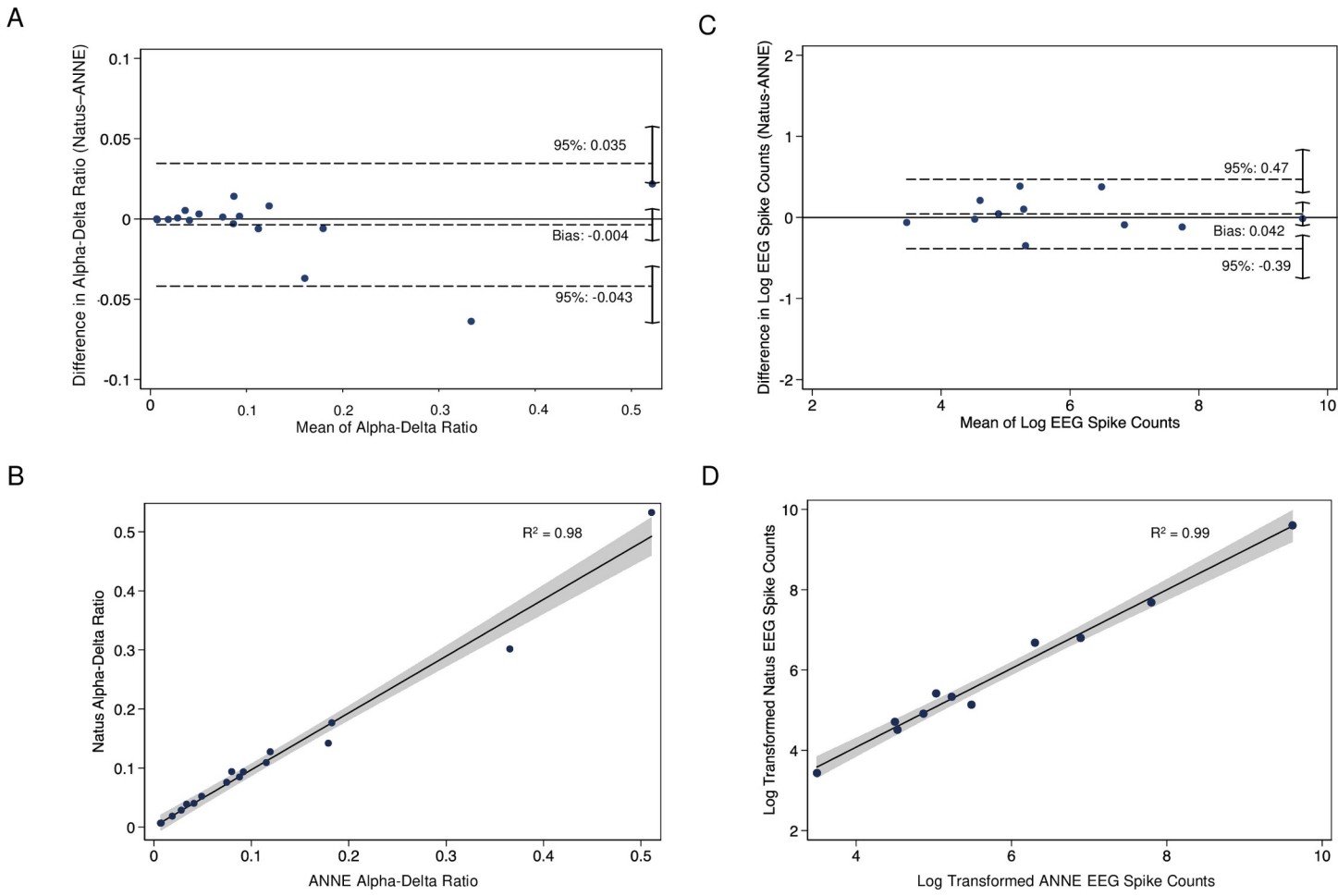

**Fig 6. Quantitative EEG comparison data to gold-standard recording system.** Data collection from study subjects with epilepsy. (A-B) Alpha-delta ratio is a validated measure of sleep. Comparison of measured alpha-delta ratios with concurrent 30 minute EEG sleep samples from ANNE EEG and gold-standard system *(A)* Bland Altman plot *(B)* Scatterplot with R-squared values. (C-D) Spikes are suggestive of epileptiform activity and are a biomarker of epilepsy. Comparison of log-transformed spike counts with concurrent 30 minute EEG samples from ANNE EEG and gold-standard system *(C)* Bland Altman plot *(D)* R-squared plot.

These findings are concordant with prior studies [26,27] that showed an increase in both these metrics for individuals with epilepsy.

In addition to successful EEG signal validation, key neurodevelopmental metrics were also investigated. For the validation cohort without epileptiform activity in sleep, Fig 8 compares their age versus distribution of EEG band power in all physiologic frequencies (delta, theta, alpha, beta and gamma) via simple linear regression analysis. A strong negative correlation was demonstrated between age and delta power, and a strong positive correlation was likewise demonstrated between age and alpha power. These findings are concordant with what has previously been reported [28,29] in neurodevelopmental studies.

A cohort of families were surveyed about their experiences with the system as part of the study. More than 95% of those surveyed (21 out of 22 respondents) stated that ANNE EEG would positively impact the frequency of physical contact with their child, with over 80% also stating that increased contact would positively impact both emotional health and emotional attachment. Overall, over 86% (19 out of 22 respondents) preferred wireless EEG to conventional EEG monitoring.

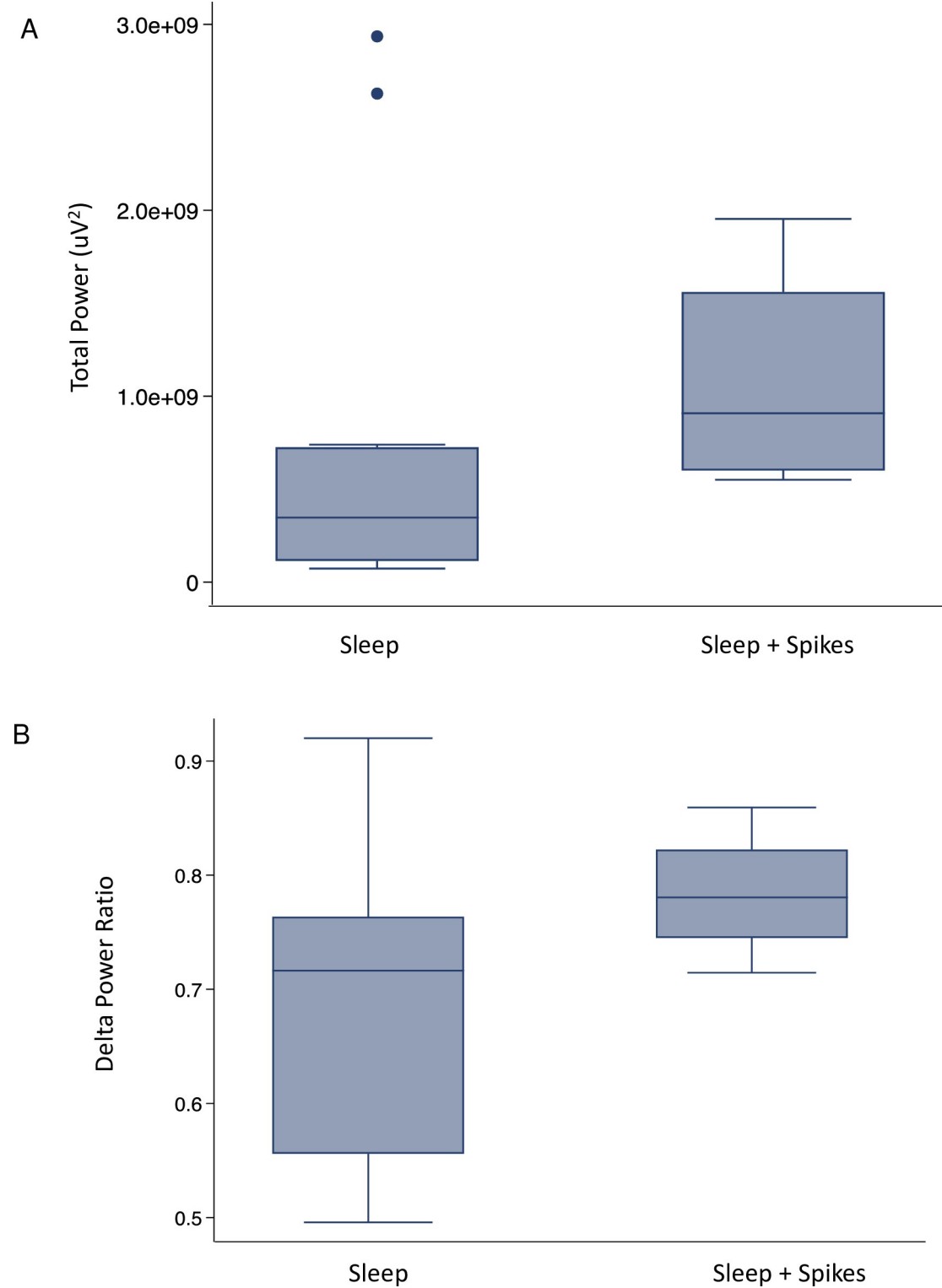

**Fig 7. Effects of epileptiform spikes on EEG power spectra.** Boxplots comparing validation study subjects in whom sleep was recorded without epileptiform spikes versus subjects with epileptiform spikes in sleep (A) Comparison of total power, demonstrating a trend towards increased power when spikes were present (B) Comparison of delta power ratio, demonstrating a trend towards an increased delta ratio when spikes were present.

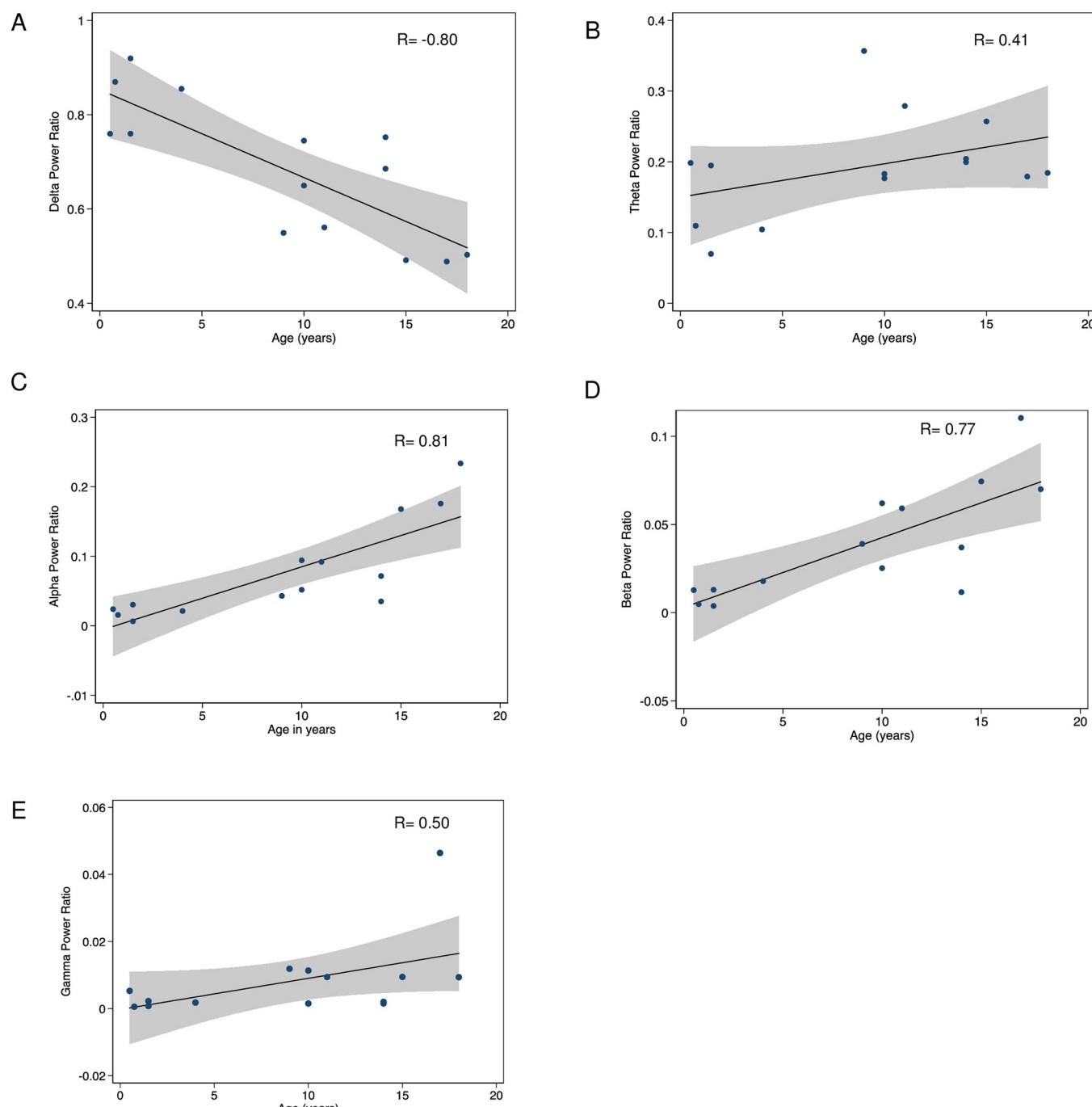

**Fig 8. Neurodevelopmental correlation of age with EEG power spectra.** Linear regression analysis comparing age and the following frequency bands: (A) Delta (0.5-4Hz), (B) Theta (4-8Hz), (C) Alpha (8-13Hz), (D) Beta (13-30Hz), (E) Gamma (30-50Hz). A strong negative correlation is demonstrated between age and delta frequency. A strong positive correlation is demonstrated between age and alpha frequency.

## Discussion

Neurodevelopment across the lifespan, in particular from birth to 10 years, is a key time period for which brain maturation occurs with long-lasting effects. Comprehensive monitoring of brain development, physical, and mental health is critical not only for the general population

but in particular for socioeconomically disadvantaged groups [30] as a means of addressing health inequalities throughout the human lifespan. EEG is a direct measurement of cerebral activity that serves as a key marker of brain development [31]. Current wired, clinical-grade EEG systems are large, cumbersome, and expensive, requiring specialized technicians to operate. While there exist some wireless EEG systems, none currently integrate with other vital sign measurements or are designed for use in neonates and children. In fact, EEG by itself is not sufficient to adequately track developmental milestones, as numerous large cohort studies [2,32] use EEG as an adjunct to other measurements. For instance, heart rate variability has been demonstrated to be reflective of attention to general stimuli [33,34] and faces [35] in infants, which is a marker of neurodevelopment. At present, there does not currently exist a wireless wearable system that measures cerebral, cardiopulmonary, and positional measurements in an integrated fashion. The ANNE system not only accomplishes this but also provides raw data outputs to best inform advanced analytics that may be helpful in predicting neurodevelopment.

Beyond neurodevelopment, the ANNE system allows for true clinical grade electroencephalography whose measurement parameters meet or exceed current standards [19], replacing standard abdominal and respiratory belts, $SpO_2$ sensor and ECG, but also adding other measurements such as SCG, temperature, accelerometry that may serve as additional variables in evaluating seizures or other conditions such as sleep apnea in high risk children, autism or movement disorders [32,36,37].

We present an innovative comprehensive wireless neurodevelopment monitoring system that adds neuronal activity measurement via EEG to previously validated chest and limb modules. Pilot studies at an academic quaternary pediatric care center demonstrate the practicality and feasibility of ANNE EEG to conduct electroencephalography studies with high levels of accuracy, validated via both quantitative and qualitative metrics, compared to the gold standard system. Furthermore, we also demonstrate the feasibility and scalability of the system for use in neonates in the LMIC setting. Previous work has already illustrated the performance and safety of the ANNE Chest and ANNE Limb sensors for cardiopulmonary monitoring in the LMIC setting [20,38,39]. The system features low-cost consumables, real-time control and streaming with widely available mobile devices (e.g. Android tablets), and fully wearable operation to allow a child to remain in their naturalistic environment. Potential use cases range from evaluating for neurologic disease at a medical clinic to brief at-home neurodevelopmental screening. An overwhelming majority of parents surveyed during studies indicated not only an overall preference for the wireless system, but also that its use would improve their children's physical and emotional health.

Our validation study has a few limitations that require further investigation prior to widespread deployment, particularly in medically underserved regions. While 30 subjects is adequate for validation of a novel EEG device validation [16] this is a relatively low number; we intend to increase this within our own institution. We also intend to expand upon the feasibility studies performed at both HIC and LMIC settings with future iterations of the system. Smaller form factors and electrode solutions that more specifically meet the needs of premature and ultra-low birth weight infants [39] would also be ideal for this vulnerable population.

While the power spectra trends displayed in Fig 7 were not statistically significant due to the small subgroup sample size, results are consistent with prior literature investigating spectral power in patients with epilepsy. We intend to investigate this trend further in future studies.

Future research investigations will validate all measurements of the multimodal monitoring system in a time-synchronized fashion. Efforts will also look to add additional measurement parameters and features based on feedback from medical care providers, such as an expanded number of electrodes and an easier-to-use wearable electrode solution.

**Table 2. Characteristics of study participants.**

|  | N = 91 |
| --- | --- |
| **Sex, n (%)** |  |
| Male | 49 (53.8) |
| Female | 42 (46.2) |
| **Neurodevelopmental age, n (%)** |  |
| Neonate (0 to 28 days) | 61 (67.0) |
| Infant (28 days to 2 years) | 6 (6.6) |
| Child (2 years to 11 years) | 16 (17.6) |
| Adolescent (12 years to 21 years) | 8 (8.8) |

## Materials and methods

### Study design and data collection

**Ethical approval.** The study and all experimental protocols were approved by the Institutional Review Boards (IRB) of Lurie Children's Hospital of Chicago (IRB 2020–3266), Columbia University (IRB-AAAT2973), and Stellenbosch University (N21/03/034_Sub-study 1173). The caregivers of all participants gave informed consent, and the studies were performed in accordance with hospital wide regulations. Written informed consent from at least one parent was given for each participant prior to sensor placement.

**Participants and informed consent.** We demonstrate performance, usability and satisfaction of ANNE EEG, in a multi-center feasibility study as well as a single center validation study. Table 2 shows demographic characteristics of all study participants, with ages ranging from birth to 18 years.

In the single center validation study, the aim was to demonstrate signal quality equivalence compared to current clinical EEG systems in a clinical cohort of 30 patients. In one patient, ANNE chest and limb modules were also added as a proof-of-concept demonstration. Table 3

**Table 3. Characteristics of validation study participants and EEG findings.**

|  | N = 30 |
| --- | --- |
| Median age, yrs (range) | 7.5 (0.5, 18) |
| **Sex, n (%)** |  |
| Male | 13 (43.3) |
| Female | 17 (56.7) |
| **Epilepsy Related Diagnosis, n (%)** |  |
| Focal epilepsy syndromes (i.e. ESES[1]) | 3 (10) |
| Genetic Syndrome (i.e. SCN1A, PRRT2 mutations) | 4 (13.3) |
| Generalized Epilepsy syndromes (i.e. CAE[2]) | 8 (26.6) |
| Structural brain abnormalities[3] | 7 (23.3) |
| Other epilepsy syndromes[4] | 8 (26.6) |
| **EEG Findings, n (%)** |  |
| Normal | 17 (56.6) |
| Diffuse slowing | 2 (6.6) |
| Epileptiform discharges | 8 (26.6) |
| Seizures | 3 (10) |

[1] Electrical Status Epilepticus of Sleep

[2] Childhood Absence Epilepsy

[3] Includes Hypoxic Ischemic Encephalopathy (HIE), meningitis, encephalitis, Tuberous Sclerosis Complex (TSC)

[4] Includes Infantile Spasms (IS) and Lennox-Gastaut Syndrome (LGS)

shows demographic characteristics of validation study participants. All the assessed participants had an epilepsy related diagnosis, ranging from genetic syndromes to structural brain abnormalities such as cerebral injury from hypoxic ischemic encephalopathy. Recruitment included patients ages 0–18 who presented to the Lurie Epilepsy Monitoring Unit (EMU) as part of a clinically indicated 4-hour EEG study. There were no other exclusion criteria. Patients were screened by a pediatric epileptologist (J.N.W.) through review of the electronic medical record.

## Data collection

At Columbia University Irving Medical Center, participants were recruited by a bilingual research assistant from the Well Baby Nursery or Neonatal Intensive Care Unit (NICU) of New York-Presbyterian Morgan Stanley Children's Hospital. Exclusion criteria included if there was a known underlying genetic condition or neonatal abstinence syndrome. Each patient underwent EEG recording for approximately 30 minutes.

At Stellenbosch University, EEG recording was performed on neonates from the Bishop Lavis Community Health Centre (CHC) born at or after 37 weeks gestational age. Exclusion criteria included delivery by caesarian section, born with congenital anomalies, diagnosed with neonatal abstinence syndrome or a 5 minute Apgar score of less than 7. EEG recording for one hour each was performed near the time of birth and then a second instance at approximately one month (27–33) days of age. For the validation study at Lurie Children's Hospital of Chicago, EEG recording with the sensor was performed concurrently with the gold-standard EEG system (Natus Xltek, Colorado USA) during clinically indicated appointments in the Lurie Epilepsy Monitoring Unit (EMU). Skin checks per hospital protocol were performed prior to each placement to ensure adequate skin integrity. Electrodes connected to the sensor were attached by trained EEG technicians using 'prime' positions of the standard 10–20 system [40], approximately 1 cm behind the corresponding electrodes connected to the Lurie system. Recording was performed for up to 4 hours, the full length of the clinically indicated hospital recording. The sensor was placed in a backpack next to the patient for the duration of the recording. EEG data from each system was downloaded to the European Data Format (EDF) and anonymized. A total of 164 hours of EEG data was recorded by the ANNE systems across all three centers.

## Statistical analysis

Bland Altman plots [46] were generated to compare the means of measured alpha-delta ratios and EEG spike counts over concurrent samples measured by the ANNE EEG and the gold-standard system. Logarithmic transformation [47] was applied to EEG spike counts as a more accurate comparison due to the wide range of spike counts across patients. These plots evaluated bias between the mean differences and estimated a 95% interval of differences between ANNE EEG and the gold standard alpha-delta ratio and EEG spike counts. Scatterplots, fitted regression lines, and correlation coefficients of the ANNE EEG-derived alpha-delta ratio and the log-transformed [47] number of EEG spikes versus the gold standard were also generated. All statistical programming and analyses were performed with STATA version 15.1 (StataCorp, College Station USA).

For the subgroup without epileptiform discharges, simple linear regression analysis was used to determine the correlation between validation subject age and power spectra for each band. The subgroup with epileptiform discharges was excluded from this specific analysis as the aim was to analyze age as a primary variable in neurodevelopment without confounding factors (such as epileptiform spikes) that would otherwise affect the EEG power spectra.

## Supporting information

**S1 Table. Validation study data.** EEG analysis timestamps, alpha-delta ratios, and automated spike counts for 30 subject validation study cohort.
(XLSX)

## Acknowledgments

The authors would like to acknowledge the technical assistance provided by Knute Martell, Jairo Chavez, and Brianna Kampmeier. We thank the Lurie Epilepsy Monitoring Unit for providing patient monitoring support.

## Author Contributions

**Conceptualization:** Jeremy N. Wong, Erin C. Conrad, Jong Yoon Lee, Sue J. Hong, Nicolò Pini, Lauren Marsillio, Erik Padilla, Olivia Gann, Ha Uk Chung, Hein J. Odendaal, William P. Fifer, Joyce Y. Wu, Shuai Xu.

**Data curation:** Jeremy N. Wong, Khrystyna Moskalyk, Mariana Vicenteno, Carlie du Plessis, Hein J. Odendaal, William P. Fifer.

**Formal analysis:** Jeremy N. Wong, Jessica R. Walter, Erin C. Conrad, Dhruv R. Seshadri, Jong Yoon Lee, Ha Uk Chung.

**Funding acquisition:** Shuai Xu.

**Investigation:** Jeremy N. Wong, Nicolò Pini, Khrystyna Moskalyk, Erik Padilla.

**Methodology:** Jeremy N. Wong, Erin C. Conrad, Dhruv R. Seshadri, Khrystyna Moskalyk, Mariana Vicenteno, Hein J. Odendaal.

**Project administration:** Jeremy N. Wong, Dhruv R. Seshadri, Carlie du Plessis, Shuai Xu.

**Resources:** Jeremy N. Wong, Jessica R. Walter, Dhruv R. Seshadri, Jong Yoon Lee, Husein Gonzalez, William Reuther, Sue J. Hong, Lauren Marsillio, Khrystyna Moskalyk, Mariana Vicenteno, Erik Padilla, Carlie du Plessis, Hein J. Odendaal, William P. Fifer, Joyce Y. Wu, Shuai Xu.

**Software:** Jeremy N. Wong, Jessica R. Walter, Erin C. Conrad, Jong Yoon Lee, William Reuther, Ha Uk Chung, Dennis Ryu, Shuai Xu.

**Supervision:** Jeremy N. Wong, Husein Gonzalez, Sue J. Hong, Lauren Marsillio, Erik Padilla, William P. Fifer, Joyce Y. Wu, Shuai Xu.

**Validation:** Jeremy N. Wong, Jessica R. Walter, Erin C. Conrad, Jong Yoon Lee, Husein Gonzalez, Olivia Gann, Ha Uk Chung, Dennis Ryu.

**Visualization:** Jeremy N. Wong, Jessica R. Walter, Erin C. Conrad, Jong Yoon Lee, William Reuther, Dennis Ryu.

**Writing – original draft:** Jeremy N. Wong, Shuai Xu.

**Writing – review & editing:** Jeremy N. Wong, Jessica R. Walter, Erin C. Conrad, Dhruv R. Seshadri, Jong Yoon Lee, Husein Gonzalez, Sue J. Hong, Nicolò Pini, Mariana Vicenteno, Erik Padilla, Joyce Y. Wu, Shuai Xu.

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
