## [Decision Letter · Decision Letter 0]

15 Dec 2022

PDIG-D-22-00177

A pediatric wireless platform for comprehensive neurological and cardiopulmonary monitoring

PLOS Digital Health

Dear Dr. Wong,

Thank you for submitting your manuscript to PLOS Digital Health. After careful consideration, we feel that it has merit but does not fully meet PLOS Digital Health's publication criteria as it currently stands. Therefore, we invite you to submit a revised version of the manuscript that addresses the points raised during the review process.

Please see editor’s comments below.

Please submit your revised manuscript within 60 days Feb 13 2023 11:59PM. If you will need more time than this to complete your revisions, please reply to this message or contact the journal office at digitalhealth@plos.org. Please include the following items when submitting your revised manuscript:

We look forward to receiving your revised manuscript.

Kind regards,

Danilo Pani, Ph.D.

Academic Editor

PLOS Digital Health

Journal Requirements:

1. Please send a completed 'Competing Interests' statement, including any COIs declared by your co-authors. If you have no competing interests to declare, please state "The authors have declared that no competing interests exist". Otherwise please declare all competing interests beginning with the statement "I have read the journal's policy and the authors of this manuscript have the following competing interests:"

3. We do not publish any copyright or trademark symbols that usually accompany proprietary names, eg ©, ®, ™ (e.g. next to drug or reagent names). Please remove all instances of trademark/copyright symbols throughout the text, including ® on pages 2, 3, 6, 7, 8, 10, 12, 13, 14, 22.

Additional Editor Comments (if provided):

The manuscript presents a commercial device assessment. As such, the novelty is limited by the impossibility to disclose the technical details of such a device. The authors should improve this, along with the statistical analysis presented in the manuscript. Is there a chance to enlarge the sample size? Please carefully address in a rebuttal letter and possibly in the manuscript the answer to the Reviewers criticisms.

Reviewers' comments:

Reviewer's Responses to Questions

**Comments to the Author**

1. Does this manuscript meet PLOS Digital Health’s publication criteria? Is the manuscript technically sound, and do the data support the conclusions? The manuscript must describe methodologically and ethically rigorous research with conclusions that are appropriately drawn based on the data presented.

Reviewer #1: Yes

Reviewer #2: Yes

2. Has the statistical analysis been performed appropriately and rigorously?

Reviewer #1: No

Reviewer #2: Yes

3. Have the authors made all data underlying the findings in their manuscript fully available (please refer to the Data Availability Statement at the start of the manuscript PDF file)?

Reviewer #1: No

Reviewer #2: Yes

4. Is the manuscript presented in an intelligible fashion and written in standard English?

Reviewer #1: Yes

Reviewer #2: Yes

5. Review Comments to the Author

Reviewer #1: In this study, the authors tried to validate a novel wireless monitoring system by showing its ability to record high-quality cerebral electric activity. ANNE® EEG which can record 16 total recording channels was used in this study. Although the idea is novel analysis results are not convincing enough to accept it as a full journal paper. A sample size of 30 subjects with varying ages is not justified to me. I have some alternative suggestions for the authors to think about if they want to resubmit the paper.

1. Increase the sample size uniformly so that there is at least 3 subjects are included for each year of neurodevelopment.

2. If not possible, then analyze other signals in addition to EEG that was presented in the current manuscript. Maybe ECG analysis including HRV could be included. Also, power spectrum analysis of EEG signals should be included to confirm the spikes. Other bands should also be investigated.

3. Include the analysis to look at the relationship between age (years) with other physiological and neurological features.

Reviewer #2: Wong et. al submitted a manuscript which introduces a novel wireless monitoring system that adds EEG measurements to an existing multimodal wearable sensor platform that has been FDA cleared to measure cardiac activity, blood oxygenation, motion, and skin temperature. Their goal was to validate the novel system by recording high quality cerebral activity via EEG during simultaneous standard EEG recording. The included 30 pediatric patients at a single academic quaternary pediatric care center as part of their pilot study. 

The authors did a wonderful job addressing the need for such a novel system, postulating its use in low to middle income countries. 

The authors detailed the components of the experimental EEG platform, ANNE ECG, which added 16 channel cerebral activity monitoring to an existing USA FDA cleared ANNE wireless monitoring platform. Based their results they were able to successfully validate their results against standard EEG recordings. Furthermore, in assessing patient/parent satisfaction, there as an overwhelming positive review. 

I commend the authors in their work which pushes to advance the field using current wireless technology. 

Points of clarification:

1/ For the 13 patients with abnormal EEG findings, could you discuss in a bit more detail how the 13 abnormal ANNE EEG varied from the gold stand? Figure 3 demonstrated almost no variation. Were the findings as consistent with patients with diffuse slowing and epileptiform discharges as well? 

2/ Practically speaking, if this system is meant market to LIMIC, what would be the assumed cost of such a system? What is the project battery life? Are any portion of the module reusable?

Minor revisions/recommendations:

Line 143. The electronics “are” (switch “is” to “are”)

6. PLOS authors have the option to publish the peer review history of their article (what does this mean?). If published, this will include your full peer review and any attached files.

**Do you want your identity to be public for this peer review?** For information about this choice, including consent withdrawal, please see our Privacy Policy.

Reviewer #1: No

Reviewer #2: No

---

## [Decision Letter · Decision Letter 1]

25 Apr 2023

PDIG-D-22-00177R1

A pediatric wireless platform for comprehensive neurological and cardiopulmonary monitoring

PLOS Digital Health

Dear Dr. Xu,

Thank you for submitting your manuscript to PLOS Digital Health. After careful consideration, we feel that it has merit but does not fully meet PLOS Digital Health's publication criteria as it currently stands. Therefore, we invite you to submit a revised version of the manuscript that addresses the points raised during the review process.

Please submit your revised manuscript within 30 days May 25 2023 11:59PM. If you will need more time than this to complete your revisions, please reply to this message or contact the journal office at digitalhealth@plos.org. Please include the following items when submitting your revised manuscript:

We look forward to receiving your revised manuscript.

Kind regards,

Danilo Pani, Ph.D.

Academic Editor

PLOS Digital Health

Journal Requirements:

3. Please send a completed 'Competing Interests' statement, including any COIs declared by your co-authors. If you have no competing interests to declare, please state "The authors have declared that no competing interests exist". Otherwise please declare all competing interests beginning with the statement "I have read the journal's policy and the authors of this manuscript have the following competing interests:"

4. Figure 2 includes an image of an identifiable person. Please provide written confirmation or release forms, signed by the subject(s) (or their parent/legally authorized guardian), giving permission to be photographed and to have their images published under our CC-BY 4.0 license. 

Otherwise, we kindly request that you remove the photograph.

Additional Editor Comments (if provided):

The manuscript was largely improved by the authors, but a comment from a Reviewer, which I also assume as relevant, was disregarded, surely for a misunderstanding on the kind of analysis. Could the authors answer this comment in a revised version of the manuscript. Moreover, I could not see Fig. 7 in the pdf of the manuscript. In the Results section, I would remove the subheading, in particular the long ones (second and third). A comment on the figures in the manuscript with some quantitative analysis would also be interesting.

Reviewers' comments:

Reviewer's Responses to Questions

**Comments to the Author**

1. If the authors have adequately addressed your comments raised in a previous round of review and you feel that this manuscript is now acceptable for publication, you may indicate that here to bypass the “Comments to the Author” section, enter your conflict of interest statement in the “Confidential to Editor” section, and submit your "Accept" recommendation.

Reviewer #1: (No Response)

2. Does this manuscript meet PLOS Digital Health’s publication criteria? Is the manuscript technically sound, and do the data support the conclusions? The manuscript must describe methodologically and ethically rigorous research with conclusions that are appropriately drawn based on the data presented.

Reviewer #1: Partly

3. Has the statistical analysis been performed appropriately and rigorously?

Reviewer #1: Yes

4. Have the authors made all data underlying the findings in their manuscript fully available (please refer to the Data Availability Statement at the start of the manuscript PDF file)?

Reviewer #1: No

5. Is the manuscript presented in an intelligible fashion and written in standard English?

Reviewer #1: Yes

6. Review Comments to the Author

Reviewer #1: Thank you for responding to my comments and revising the manuscript. However my comment 3 is still not addressed. Please inlcude a figure showing correlation of EEG band power (alpha, beta, theta and gamma) with ages (years) of the participants. Otherwise I would not be able to understand how neurodevelopment can be explained by your study design. 

Also I could not find analysis results of additional data in the figures.

7. PLOS authors have the option to publish the peer review history of their article (what does this mean?). If published, this will include your full peer review and any attached files.

**Do you want your identity to be public for this peer review?** For information about this choice, including consent withdrawal, please see our Privacy Policy. 

Reviewer #1: No

---

## [Editor Report · Decision Letter 2]

1 Jun 2023

A pediatric wireless platform for comprehensive neurological and cardiopulmonary monitoring

PDIG-D-22-00177R2

Dear Dr. Xu,

We are pleased to inform you that your manuscript 'A pediatric wireless platform for comprehensive neurological and cardiopulmonary monitoring' has been provisionally accepted for publication in PLOS Digital Health.

Best regards,

Danilo Pani, Ph.D.

Academic Editor

PLOS Digital Health

The authors addressed all the points. I suggest, when submitting the final version for the production, to mind the number of significant digits in the added table.